# N-Acetylcysteine Attenuates Sepsis-Induced Muscle Atrophy by Downregulating Endoplasmic Reticulum Stress

**DOI:** 10.3390/biomedicines12040902

**Published:** 2024-04-18

**Authors:** Renyu Chen, Yingfang Zheng, Chenchen Zhou, Hongkai Dai, Yurou Wang, Yun Chu, Jinlong Luo

**Affiliations:** 1Department of Emergency Medicine, Tongji Hospital, Tongji Medical College, Huazhong University of Science and Technology, Wuhan 430030, China; m202176293@hust.edu.cn (R.C.); d202081966@hust.edu.cn (Y.Z.); m202076352@hust.edu.cn (C.Z.); d202382348@hust.edu.cn (H.D.); m202176452@hust.edu.cn (Y.W.); m202276297@hust.edu.cn (Y.C.); 2Department of Critical Care Medicine, Tongji Hospital, Tongji Medical College, Huazhong University of Science and Technology, Wuhan 430030, China

**Keywords:** N-acetylcysteine, endoplasmic reticulum stress, sepsis, muscle atrophy

## Abstract

(1) Background: Sepsis-induced muscle atrophy is characterized by a loss of muscle mass and function which leads to decreased quality of life and worsens the long-term prognosis of patients. N-acetylcysteine (NAC) has powerful antioxidant and anti-inflammatory properties, and it relieves muscle wasting caused by several diseases, whereas its effect on sepsis-induced muscle atrophy has not been reported. The present study investigated the effect of NAC on sepsis-induced muscle atrophy and its possible mechanisms. (2) Methods: The effect of NAC on sepsis-induced muscle atrophy was assessed in vivo and in vitro using cecal ligation and puncture-operated (CLP) C57BL/6 mice and LPS-treated C2C12 myotubes. We used immunofluorescence staining to analyze changes in the cross-sectional area (CSA) of myofibers in mice and the myotube diameter of C2C12. Protein expressions were analyzed by Western blotting. (3) Results: In the septic mice, the atrophic response manifested as a reduction in skeletal muscle weight and myofiber cross-sectional area, which is mediated by muscle-specific ubiquitin ligases—muscle atrophy F-box (MAFbx)/Atrogin-1 and muscle ring finger 1 (MuRF1). NAC alleviated sepsis-induced skeletal muscle wasting and LPS-induced C2C12 myotube atrophy. Meanwhile, NAC inhibited the sepsis-induced activation of the endoplasmic reticulum (ER) stress signaling pathway. Furthermore, using 4-Phenylbutyric acid (4-PBA) to inhibit ER stress in LPS-treated C2C12 myotubes could partly abrogate the anti-muscle-atrophy effect of NAC. Finally, NAC alleviated myotube atrophy induced by the ER stress agonist Thapsigargin (Thap). (4) Conclusions: NAC can attenuate sepsis-induced muscle atrophy, which may be related to downregulating ER stress.

## 1. Introduction

Skeletal muscle is a major protein reservoir of the human body and is critical for energy, metabolism, and body function [1]. Skeletal muscle atrophy is characterized by loss of muscle mass, weakened function, and reduced myofiber size [2,3]. In pathological conditions, protein synthesis and degradation in skeletal muscle are unbalanced, causing degradation of skeletal muscle structural proteins. Severe skeletal muscle atrophy may even leave the patient incapacitated, compromising their quality of life [4]. Sepsis-induced muscle atrophy influences the prognosis of septic patients and prolongs disease recovery time [5]. Previous studies have demonstrated that the ubiquitin–proteasome system (UPS) plays a vital role in sepsis-induced muscle proteolysis. Atrogin-1 and MuRF1 are two important ubiquitin ligases that mediate the degradation of muscle proteins [2,6]. Several studies have alleviated muscle atrophy by inhibiting the expression of Atrogin-1 and MuRF1 [7,8].

Endoplasmic reticulum (ER) stress is a response to the accumulation of unfolded or misfolded proteins in the lumen of the endoplasmic reticulum. An overabundance of unfolded or misfolded protein activates the unfolded protein response (UPR) [9,10]. Skeletal muscle has a rich sarcoplasmic reticulum and ER stress-induced UPRs are activated under multiple conditions in skeletal muscle [11,12]. In order to restore homeostasis, the UPR involves the activation of three transmembrane sensors: protein kinase R (PKR)-like endoplasmic reticulum kinase (PERK), inositol-requiring protein (IRE) 1α, and activating transcription factor-6 (ATF6), which bind to glucose-regulated protein (GRP78) in normal physiological conditions [13,14]. GRP78 dissociates from them to bind misfolded proteins upon ER stress, leading to a cascade of signals and the activation of transcription factors such as CCAAT enhancer binding protein (C/EBP) homologous protein (CHOP) and growth arrest and DNA damage-inducible protein (GADD34) [15]. In a former research study [16], ER stress is activated and involved in disuse-induced muscle atrophy. Additionally, our previous study [17] reported that ER stress promotes sepsis-induced muscle atrophy. The upregulation of biomarkers of ER stress follows an analogous trend to that of atrophy-related genes, which reveals a correlation between ER stress and sepsis-induced muscle atrophy. Hence, ER stress is anticipated to be a promising new candidate target for the treatment of sepsis-induced muscle atrophy.

N-acetylcysteine (NAC), a precursor of glutathione (GSH), has anti-inflammatory, antioxidant, and cytoprotection properties, and it is deacetylated intracellularly to form L-cysteine and acts as potential H2S-releasing donor [18,19,20]. NAC protects against sepsis-associated encephalopathy, septic acute lung injury, and septic acute kidney injury, and its mechanism of action is related to anti-inflammatory, antioxidant, and anti-apoptotic functions [21,22,23]. Recently, a study found that NAC prevented skeletal muscle atrophy in diabetes [24]. In addition, NAC alleviates diaphragm contractile dysfunction due to mechanical ventilation and reduces ubiquitin ligase expression [25]. Olivier et al. demonstrated a protective effect of N-acetylcysteine against bupivacaine-induced sarcoplasmic/endoplasmic reticulum stress in primary human skeletal muscle cells [26]. However, the effect of NAC on sepsis-induced muscle atrophy is unclear. Thus, this study aims to investigate the effect of NAC on sepsis-induced muscle atrophy and its possible mechanisms.

## 2. Materials and Methods

### 2.1. Materials

N-acetylcysteine was purchased from Kang Enbei Biopharmaceutical Co., Ltd. (H20057334, Hangzhou, Zhejiang, China). Lipopolysaccharide (LPS) from Escherichia coli (O111:B4) was purchased from Sigma-Aldrich (St. Louis, MO, USA). All materials used for SDS-PAGE were purchased from Servicebio (Wuhan, China), and a polyvinylidene difluoride (PVDF) membrane was attained from Millipore (Burlington, MA, USA). 4-Phenylbutyric acid (4-PBA, CAS No.: 1821-12-1) and Thapsigargin (Thap, CAS No.: 67526-95-8) were supplied by Medchemexpress (Monmouth Junction, NJ, USA). All antibodies used were as described in our previous article [17].

### 2.2. Animals and Drug Treatment

The Animal Ethics Committee of Tongji Hospital approved all animal experiments conducted in this study (TJH-202304014). Cecum ligation and perforation (CLP) was used to establish a sepsis model [27,28]. Initially, we anesthetized mice with sodium pentobarbital solution (1%, 50 mg/kg i.p.). After shaving the abdominal hair of the mice, the abdominal skin of the mice was disinfected with 75% alcohol. An abdominal midline incision was made to expose the cecum. Subsequently, the cecum was ligated at the distal 1/2 and perforated with a 21 G needle. Next, the cecum was placed back into the abdominal cavity and the abdomen was sutured closed. Preheated saline (37 °C, 50 mL/kg s.c.) and buprenorphine (0.05 mg/kg s.c.) were used for fluid resuscitation and postoperative analgesia. All operations were gentle. Mice in the sham-operation group were operated on in the same way except for ligation and perforation. C57BL/6J mice (male, 8 weeks, 22–25 g) were randomly assigned into four groups: the sham-operation group (*n* = 4), the CLP group (*n* = 8), the CLP + NAC (0.5 g/kg) group (*n* = 8), and the CLP + NAC (1 g/kg) group (*n* = 8). NAC was prepared at two concentrations with saline and administered by gavage according to the specified concentrations in the CLP + NAC (0.5 mg/kg) group and CLP + NAC (1 mg/kg) group preoperatively at 1 h, postoperatively at 24 h, and postoperatively at 48 h. Mice in the sham-operation group and CLP group were administrated the same volume of saline by gavage. Mice were killed under anesthesia 72 h after surgery, and the muscles (tibialis anterior, gastrocnemius) were harvested, quickly frozen in liquid nitrogen, and stored at −80 °C.

### 2.3. Cell Culture and Drug Treatment

*Mouse* C2C12 myoblasts were purchased from Cellverse Bioscience Technology Co., Ltd. (Shanghai, China). The C2C12 myoblasts were cultured prior to differentiation in Dulbecco’s modified Eagle’s medium (DMEM) with 10% FBS and 100 units/mL of penicillin and streptomycin (PS) at 37 °C and 5% CO_2_. After reaching 80% confluence, the myoblasts were induced to differentiate into myotubes using a differentiation medium (DMEM supplemented with 2% horse serum and 100 units/mL of PS) for 4 days. The medium was changed every 2 days. To investigate the impact of NAC on the LPS-induced myotube atrophy, the C2C12 myotubes were divided into four groups: the control group, the LPS-treated group, the LPS + NAC (0.1 mM) group, and the LPS + NAC (1 mM) group. NAC was dissolved in phosphate-buffered saline (PBS). Prior to the addition of LPS (10 μg/mL), the myotubes were pretreated with NAC for 2 h, using the corresponding concentration. The control group and LPS group were both added with an equal volume of PBS.

### 2.4. Cell Viability

The CCK-8 assay kit (Beyotime Biotechnology, Shanghai, China) was used to test cell viability. C2C12 cells were seeded in a 96-well plate at a density of 1 × 10^4^ cells per well. CCK-8 (10Μl) solution was added to the wells for 1 h after 24 h of NAC (0.1 mM, 0.5 mM, 1 mM, 5 mM) treatment and the absorbance was measured at 450 nm.

### 2.5. Western Blot

C2C12 myotubes and TA muscles were washed twice with 4 °C PBS buffer. Then, they were lysed for 10 min in radioimmunoprecipitation assay (RIPA) lysis buffer supplemented with protease and phosphatase inhibitor cocktails (Medchemexpress, Monmouth Junction, NJ, USA). The protein homogenate was centrifuged at 4 °C and 12,000× *g* rpm for 15 min, followed by using a BCA protein assay kit to determine the protein concentration. Equal amounts of total protein (40 μg of protein per lane) were separated by SDS-PAGE on 10% polyacrylamide gels and transferred to PVDF membranes. The membranes were blocked with 5% skim milk at room temperature for 1 h. After incubating with primary antibodies at 4 °C overnight, the membranes were washed 3 times in Tris-buffered saline with Tween 20 (TBST) and incubated with the secondary antibody. The protein bands were detected with a high-sensitivity ECL kit (Medchemexpress, Monmouth Junction, NJ, USA).

### 2.6. Immunofluorescence Staining of Myofibers

The TA muscle was placed with an OCT compound (Sakura, Tokyo, Japan), flash-frozen in liquid nitrogen, and cut into 8-μm-thick muscle slices on a cryostat (Leica CM1950, Wetzlar, Germany). The sections were fixed with 4% paraformaldehyde for 10 min, permeabilized with 0.5% TritonX-100, and blocked with 5% BSA for 1 h at room temperature. Subsequently, the sections were incubated with anti-laminin alpha-2 (1:200) at 4 °C overnight. The sections were washed four times with phosphate-buffered saline with Tween 20 (PBST) and then incubated with the fluorescent secondary antibody, Goat Anti-rat IgG H&L (ZF-0315, ZSGB Biopharmaceutical Co., Ltd., Beijing, China), in a dark cassette. We washed the sections with PBST four more times and stained the nuclei with DAPI (Servicebio, Wuhan, China) for 5 min. Eventually, the cross-sectional area (CSA) of the TA muscles was photographed with an OLYMPUS BX51 (Tokyo, Japan) and analyzed (three randomly selected images per mouse) using the ImageJ software (ImageJ 1.53t, Wayne Rasband, Bethesda, MD, USA).

### 2.7. Cell Immunofluorescence

C2C12 myotube diameters were measured by immunofluorescence. The myotubes were washed three times with PBS, and then fixed with 4% paraformaldehyde for 15 min and permeabilized with 0.5% TritonX-100. The cells were blocked with 5% BSA for 1 h to reduce non-specific binding. After incubating with anti-MyHC (1:150) overnight at 4 °C, the myotubes were washed with PBS and incubated with a specific fluorescent secondary antibody: Goat Anti-mouse IgG H&L (ZF-0312, ZSGB Biopharmaceutical Co., Ltd., Beijing, China). The nuclei were stained with DAPI for 5 min. Images were observed and captured with an OLYMPUS IX71 (Tokyo, Japan). Ultimately, the images were analyzed with the ImageJ software (ImageJ 1.53t, Wayne Rasband, Bethesda, MD, USA). Images were randomly selected from each group and the diameter of 100 myotubes was counted.

### 2.8. Statistical Analysis

The mean ± standard deviation (SD) was utilized to express all values. Statistical analysis was performed using GraphPad Prism 9.5.1 (GraphPad Software, San Diego, CA, USA). *p* values were calculated using Student’s *t*-test or a one-way analysis of variance (ANOVA). A value of *p* < 0.05 represented statistical significance.

## 3. Results

### 3.1. N-Acetylcysteine Prevents Sepsis-Induced Muscle Atrophy In Vivo

The chemical structure of N-acetylcysteine is shown as Figure 1A. We investigated the protective effect of NAC on sepsis-induced muscle atrophy in a sepsis model with perforated cecum ligation (CLP). NAC rescued body weight loss caused by sepsis (Figure 1B). At 72 h after surgery, we dissected the tibialis anterior (TA) and gastrocnemius muscles and weighed them. Mice in the CLP group had significantly lower muscle weights compared to the sham group. Nevertheless, NAC treatment was able to mitigate the decline in muscle weights (Figure 1D,E). Loss of muscle mass was often connected to the size and number of myofibers. Therefore, we performed immunofluorescence staining on muscle sections of the tibialis anterior muscle to measure cross-sectional area (CSA). We observed a reduction in the CSA of TA muscle in the CLP group compared to the sham group, while the CSA of TA muscle in NAC treatment group recovered (Figure 1F,G). In addition, quantification of myofiber size revealed a leftward shift in the CSA distribution of myofiber in the CLP group. Meanwhile, in comparison to the CLP group, the CSA distribution of myofibers in the NAC-treated group was shifted rightwards (Figure 1H). These results suggest that NAC administration ameliorated myofiber atrophy. Previous studies have shown that sepsis-induced muscle atrophy is mainly mediated by the ubiquitin–proteasome system (UPS). To interrogate the influence of NAC on sepsis-induced muscle atrophy at the molecular level of proteins, we used two E3 ubiquitin ligases that were almost activated during muscle atrophy: Atrogin-1 and MuRF1 [3,29]. We also examined skeletal muscle structural protein myosin heavy chain (MyHC) and poly-ubiquitinated proteins levels. Sepsis increased the protein expression of Atrogin-1, MuRF1, and poly-ubiquitinated proteins while downregulating the expression of MyHC. Nevertheless, the expression of Atrogin-1, MuRF1, and poly-ubiquitinated protein was decreased in the NAC-treated group. In addition, the expression of MyHC was restored (Figure 1I–M). The preceding results suggest that NAC suppressed the UPS and reduced Atrogin-1 and MuRF1-mediated protein degradation to protect against sepsis-induced muscle atrophy.

### 3.2. N-Acetylcysteine Attenuates LPS-Induced Myotube Atrophy In Vitro

To evaluate the effect of NAC on LPS-induced myotube atrophy, we followed the procedure in Figure 2A to process the cells. The CCK-8 assay was used to assess the cell viability of C2C12 cells with different concentrations of NAC (0.1 mM, 0.5 mM, 1 mM, 5 mM) for 24 h. Among them, 0.1 mM, 0.5 mM, and 1 mM of NAC cultured with cells for 24 h had no significant effect on the viability of cells compared to the control group (Figure 2B). In order to exclude the effect of NAC itself on myotubular cells, we also performed immunofluorescence staining of myotubular cells treated with different concentrations of NAC to evaluate the effect of NAC on the diameter of myotubular cells. The treatment of myotube cells with NAC alone revealed no statistical difference in myotube diameter (Supplemental Appendix A), which suggests that NAC functioned only in the context of LPS treatment. We took 0.1 mM and 1 mM of NAC as low- and high-concentration drug treatments, respectively. LPS-treated myotubes were significantly decreased in diameter, whereas NAC-pretreated myotubes recovered (Figure 2C,D). Similarly, we examined the expression of the above-mentioned proteins in C2C12 myotubes. In the LPS treatment group, the expression of Atrogin-1, MuRF1, and poly-ubiquitinated protein was increased and the protein level of MyHC showed a declining trend. However, administration of NAC reversed the expression of the above proteins in LPS-induced myotubes (Figure 2E–I). The current results suggest that NAC pretreatment similarly inhibited UPS activation and ameliorated LPS-induced myotube atrophy in vitro.

### 3.3. N-Acetylcysteine Inhibits the Activation of Sepsis-Induced Endoplasmic Reticulum Stress

Our published study demonstrates that multiple markers of ER stress are activated in muscle tissue in sepsis, such as glucose-regulated protein (GRP78), inositol-requiring protein 1 (IRE1), spliced X box binding protein 1 (sXBP1), activating transcription factor-6 (ATF6), eukaryotic translation initiation factor 2α (eIF2α), growth arrest and DNA damage-inducible protein (GADD34), and CCAAT enhancer binding protein (C/EBP) homologous protein (CHOP) [17]. In this study, we chose several ER stress biomarkers to assess the level of ER stress during the experiments. IRE1 is one of transmembrane sensors of ER stress. It is auto-phosphorylated during ER stress, along with promoting splicing of the 26-base intron of X box binding protein 1 (XBP1mRNA) through its endonuclease activity, and sXBP1 acts as an active transcription factor to participate in subsequent signaling. PERK is activated by auto-phosphorylation as well, leading to a cascade of signals, including the activation of eIF2α and CHOP. The PERK/eIF2α signaling axis also induces the expression of GADD34 [30,31]. Initially, we found that sepsis induces the expression of GRP78, IRE1, sXBP1, eIF2α, GADD34, and CHOP. But with NAC administration in septic mice, the expression of the above biomarkers of ER stress in muscles significantly decreased (Figure 3A–G). Likewise, analogous experimental results were observed in LPS-induced myotube atrophy. LPS treatment activated ER stress and elevated ER stress marker protein levels versus decreased ER stress in the NAC pretreated group (Figure 3H–N). Consequently, these results support the finding that NAC inhibited the activation of sepsis-induced ER stress.

### 3.4. The Protective Effect of NAC on Sepsis-Induced Muscle Atrophy Is Partly Mediated by ER Stress

We used the ER stress pathway inhibitor, 4-Phenylbutyric acid (4-PBA), to verify that the protective effect of NAC on sepsis-induced muscle atrophy was mediated by ER stress. The cell procedure and processing are shown in Figure 4A. 4-PBA was utilized to inhibit ER stress at concentrations of 0.5mM, 1mM, and 2 mM. The effect of 4-PBA is illustrated in Figure 4B. The concentration of 2 mM of 4-PBA was selected as the target concentration. C2C12 myotubes were divided into five groups—the control group, LPS group, LPS + NAC group, LPS + 4-PBA group, and LPS + NAC + 4-PBA group—to assess the protective effect of NAC on LPS-induced myotube atrophy. The myotube diameter increased in both the LPS + NAC group and the LPS + 4-PBA group compared to the LPS group (Figure 4C,D). Concomitant administration of NAC and 4-PBA also restored the myotube diameter in LPS-treated myotubes but was not statistically different from 4-PBA alone (Western blot as shown in Figure 4E). Pretreatment of NAC or 4-PBA both reduced the LPS-induced expression of Atrogin-1, MuRF1, and poly-ubiquitinated proteins and reversed decreased MyHC protein expression. Yet there was no statistically significant difference in protein expression between the LPS + 4-PBA group and the LPS + NAC + 4-PBA group (Figure 4F–I). Accordingly, we considered that the protective effect of NAC on sepsis-induced muscle atrophy is partly mediated by ER stress.

### 3.5. N-Acetylcysteine Attenuates ER Stress-Induced Myotube Atrophy

This part of the experiment is as shown in Figure 5A. We used various concentrations (0.01 μM, 0.1 μM, 0.5 μM) of the ER stress agonist Thapsigargin to provoke ER stress (Figure 5B). We chose 0.5 μM as the concentration of Thapsigargin for the following experiments. C2C12 myotubes were separated into four groups: the control group, Thapsigargin group, Thapsigargin + NAC (0.1 mM) group, and Thapsigargin + NAC (1 mM) group. The myotube diameter was decreased in the Thapsigargin-treated group; however, it was restored in the NAC-treated group (Figure 5C,D). Thapsigargin treatment increased the expression of Atrogin-1, MuRF1, and poly-ubiquitinated protein and decreased MyHC expression (Figure 5E–I). These results are supportive of ER stress promoting myotube atrophy and NAC attenuating ER stress-induced myotube atrophy.

## 4. Discussion

Sepsis-induced muscle atrophy increases the burden on society, prolongs hospitalization, affects patients’ quality of life, and is associated with increased mortality [5,32]. Despite emerging research on the mechanisms of sepsis-induced muscle atrophy, there is a dearth of drugs that can be used for clinical therapy. NAC has powerful anti-inflammatory and antioxidant properties and has therapeutic and protective effects on muscle wasting from a variety of causes, but its role in sepsis-induced muscle atrophy has not been reported. Our study revealed that NAC can attenuate sepsis-induced muscle wasting in vivo and LPS-induced myotube atrophy in vitro, which is associated with the regulation of the ubiquitin–proteasome system (UPS). Apart from this, we found that NAC inhibited the activation of sepsis-induced ER stress, and inhibiting the ER stress pathway blocked the anti-muscle-atrophy effect of NAC. In addition, NAC could alleviate ER stress-induced myotube atrophy. These findings suggest that the anti-muscle-atrophy effect of NAC on sepsis was associated with suppressing the ER stress pathway.

Sepsis-induced muscle atrophy is mainly due to an imbalance between protein synthesis and degradation, and the ubiquitin–protease system (UPS) is widely recognized as an influential pathway mediating protein degradation in sepsis-induced muscle atrophy [33]. In sepsis, the ubiquitination of proteins involves the participation of E1 ubiquitin-activating enzymes, E2 ubiquitin-conjugating enzymes, and E3 ubiquitin-protein ligases, and subsequently, ubiquitinated proteins are recognized, bound, and degraded by the 26S proteasome [34]. The muscle-specific E3 ubiquitin ligases Atrogin-1 and MuRF1 have been reported in sepsis-induced muscle atrophy [35,36], which we chose as evaluators to detect the effect of NAC on sepsis-induced muscle atrophy. Our results showed that NAC was able to reduce the sepsis-induced expression of poly-ubiquitinated proteins, Atrogin-1 and MuRF1, thereby decreasing weight and muscle mass loss and restoring the cross-sectional area (CSA) of TA muscle and MyHC expression. Similar results were reproduced in LPS-induced myotubes. NAC reinstated myotube diameter and MyHC expression through inhibiting Atrogin-1 and MuRF1. From these, we concluded that NAC attenuated sepsis-induced muscle atrophy by suppressing UPS activation and decreasing protein degradation.

ER stress is activated in response to different stimuli, among which the activation of ER by inflammation and proinflammatory factors has been widely studied. It has been reported that TLR4 and TLR2 specifically activate the ER stress sensor kinases IRE1 and XBP1 [37]. Delmotte et al. found that TNF-α activates the ER stress pathway in human airway smooth muscle cells [38]. Sepsis is an uncontrolled systemic inflammatory response; inflammatory mediators such as TNF-α and IL-6 are elevated in sepsis, and it has been described that ER stress is dramatically induced in sepsis [39,40,41]. Inflammation is considered to be the major predisposing factor for skeletal muscle atrophy, which disturbs the homeostasis of the ER [42,43]. Notably, ER stress also affects the inflammatory process in turn [30]. Therefore, we examined ER stress levels in septic mice and LPS-induced myotubes after NAC treatment and found that ER stress biomarkers followed the same trend as Atrogin-1 and MuRF1, with NAC decreasing their expression levels. It has been reported that NAC inhibits TNF-α or other proinflammatory cytokine-induced nuclear factor kappa B (NF-κB) activation, which is central to the regulation and expression of inflammation [18,44]. We assumed that NAC inhibited sepsis-induced ER stress in relation to its potent anti-inflammatory effect. Due to the multiple interactions between the ER stress pathway and the inflammatory response, the underlying mechanisms by which NAC plays a role in sepsis-induced muscle atrophy deserve further investigation. As mentioned previously, NAC is a potential H2S-releasing donor [18,23]. Ferlito et al. elucidated that H2S inhibits CHOP expression and improves survival in septic mice [45]. Therefore, we are not ruling out the possibility that H2S may be indirectly involved in regulating ER stress. In addition, NAC prevented ER stress induced by valproic acid + acetaminophen co-administration, which may be related to the regulation of GSH levels by NAC [46].

We used inhibitors of the ER stress pathway to further explore the role of NAC inhibiting sepsis-induced ER stress and muscle atrophy. 4-Phenylbutyric acid (4-PBA) was confirmed to be able to inhibit ER stress induced by a variety of diseases [47,48]. As mentioned previously, both the LPS + NAC group and the LPS + 4-PBA group reduced LPS-induced Atrogin-1 and MuRF1 expression, whereas when LPS-induced myotubes were pretreated with NAC and 4-PBA, atrophy protein expression and myotube diameter in the LPS + NAC + 4-PBA group were not statistically different from the LPS + 4-PBA group, which meant that the anti-muscle-atrophy effect of NAC acted partly through ER stress. Next, considering that NAC is a powerful antioxidant, to reduce the interference of oxidative stress, we induced C2C12 myotube atrophy using an ER stress pathway agonist to evaluate effect of NAC. Thapsigargin (Thap) is the sarco/endoplasmic reticulum Ca2 + -ATPase pump inhibitor and triggers ER stress [49]. We found that Thap can induce myotubes to atrophy, and NAC reduced the expression of Thap-induced atrophy-related proteins and restored the reduction in myotube diameter, suggesting that NAC inhibited ER stress-induced myotube atrophy.

Since sepsis-induced muscle atrophy has serious clinical consequences, it is urgent to find effective drugs to prevent and treat it. With the progressive study of ER stress in sepsis-induced muscle atrophy, ER stress has emerged as a promising new target for the treatment of sepsis-induced muscle atrophy. In this study, we found that NAC attenuated sepsis-induced muscle atrophy by downregulating ER stress. Thus, our results provide a rationale for NAC treatment and prevention of sepsis-induced muscle atrophy.

## Figures and Tables

**Figure 1 biomedicines-12-00902-f001:**
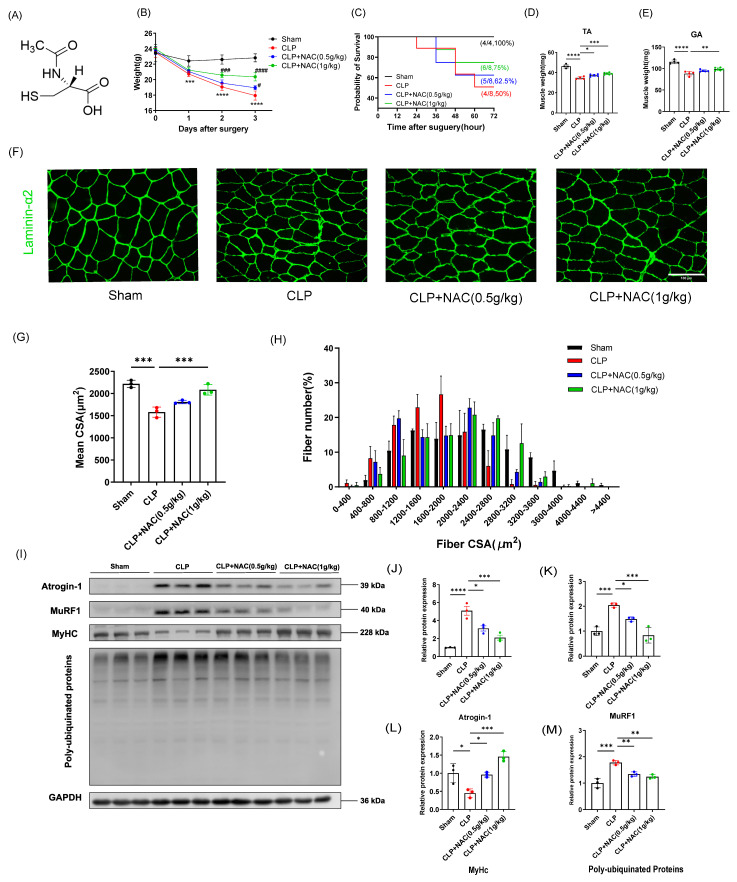
N–acetylcysteine prevents sepsis-induced muscle atrophy in vivo. (**A**) Chemical structure of N–acetylcysteine. (**B**) Body weight changes after surgery in each group (*n* = 4–6). (**C**) Probability of survival after surgery. (**D**,**E**) Muscle weight of TA and GA (*n* = 4–6). (**F**) Representative immunofluorescent staining of myofiber cross-section of TA. The scale bar represents 100 μm. (**G**) Average CSA for TA of each group. (**H**) Quantification of fiber size measured by CSA (*n* = 3). (**I**–**M**) Western blot and quantitative analysis of Atrogin-1, MuRF1, MyHC, and poly-ubiquitinated proteins (*n* = 3). GAPDH is used as internal control. TA: tibialis anterior muscle; GA: gastrocnemius muscle; CSA: cross-sectional area. Black: Sham-operation group; Red: CLP group; Blue: CLP + NAC(0.5g/kg) group; Green: CLP + NAC(1g/kg) group. Differences are considered significant at levels of * *p* < 0.05, ** *p* < 0.01, *** *p* < 0.001, and **** *p* < 0.0001 versus the sham group. Differences are considered significantly at levels of # *p* < 0.05, ### *p* < 0.001, and #### *p* < 0.0001 versus the CLP group.

**Figure 2 biomedicines-12-00902-f002:**
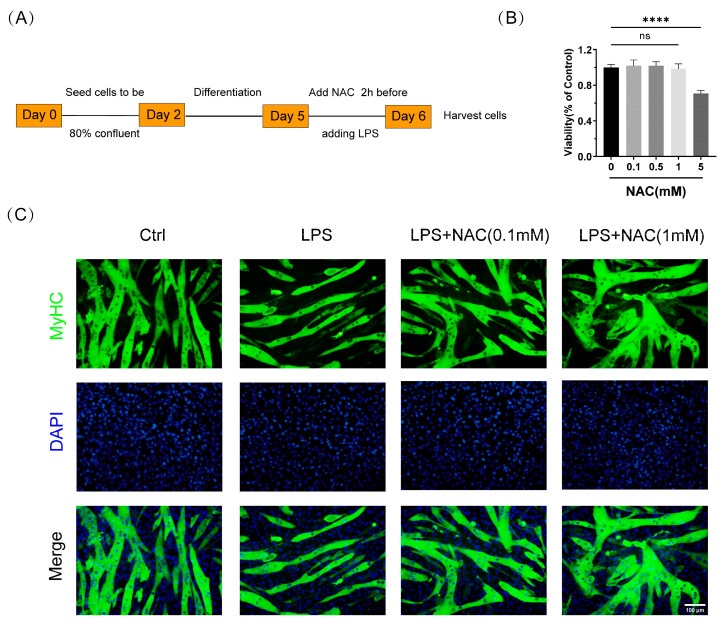
N–acetylcysteine attenuates LPS-induced myotube atrophy in vitro. (**A**) The protocol of cell experiments. (**B**) C2C12 myotubes were treated with different concentrations (0.1 mM, 0.5 mM, 1 mM, 5 mM) of NAC for 24 h. Cell viability was detected by a CCK-8 assay. (**C**) Representative images of immunofluorescence of MyHC in the four groups (*n* = 3). Scale bar = 100 μm. (**D**) Measurement of myotube diameters after stimulation of LPS with or without the addition of NAC. (**E**) The expression levels of Atrogin-1, MuRF1, MyHC, and poly-ubiquitinated proteins were measured in LPS-induced myotubes with or without NAC treatment by Western blot analysis (*n* = 3). (**F**–**I**) Quantification of the proteins indicated above. Ctrl: control group; LPS: lipopolysaccharide; NAC:N–acetylcysteine. Black: control group; Red: LPS group; Blue: LPS + NAC (0.1 mM) group; Green: LPS + NAC (1 mM) group. * *p* < 0.05, ** *p* < 0.01, *** *p* < 0.001, and **** *p* < 0.0001 represent a significant difference. ns = not statistically significant.

**Figure 3 biomedicines-12-00902-f003:**
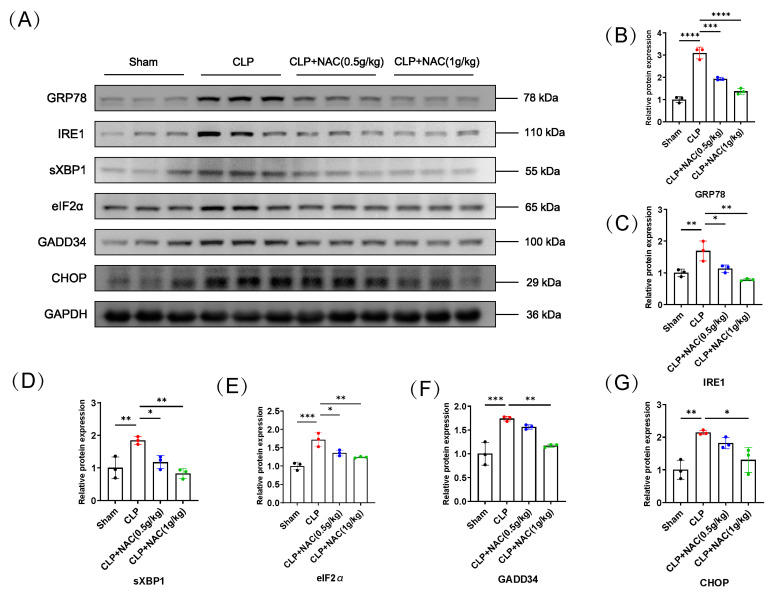
N–acetylcysteine inhibits endoplasmic reticulum stress in sepsis. (**A**) Western blot assay for ER stress biomarkers: GRP78, IRE1, sXBP1, eIF2α, GADD34, and CHOP in septic mice (*n* = 3). (**B**–**G**) Quantifications of (**A**). (**H**) Expression of the aforementioned ER stress proteins in C2C12 myotubes were assessed by Western blot (*n* = 3). (**I**–**N**) Quantifications of (**H**) are shown. For (**B**–**G**), Black: Sham-operation group; Red: CLP group; Blue: CLP + NAC (0.5 g/kg) group; Green: CLP + NAC (1 g/kg) group. For (**I**–**N**), Black: control group; Red: LPS group; Blue: LPS + NAC (0.1 mM) group; Green: LPS + NAC (1 mM) group. * *p* < 0.05, ** *p* < 0.01, *** *p* < 0.001, and **** *p* < 0.0001 represent a significant difference.

**Figure 4 biomedicines-12-00902-f004:**
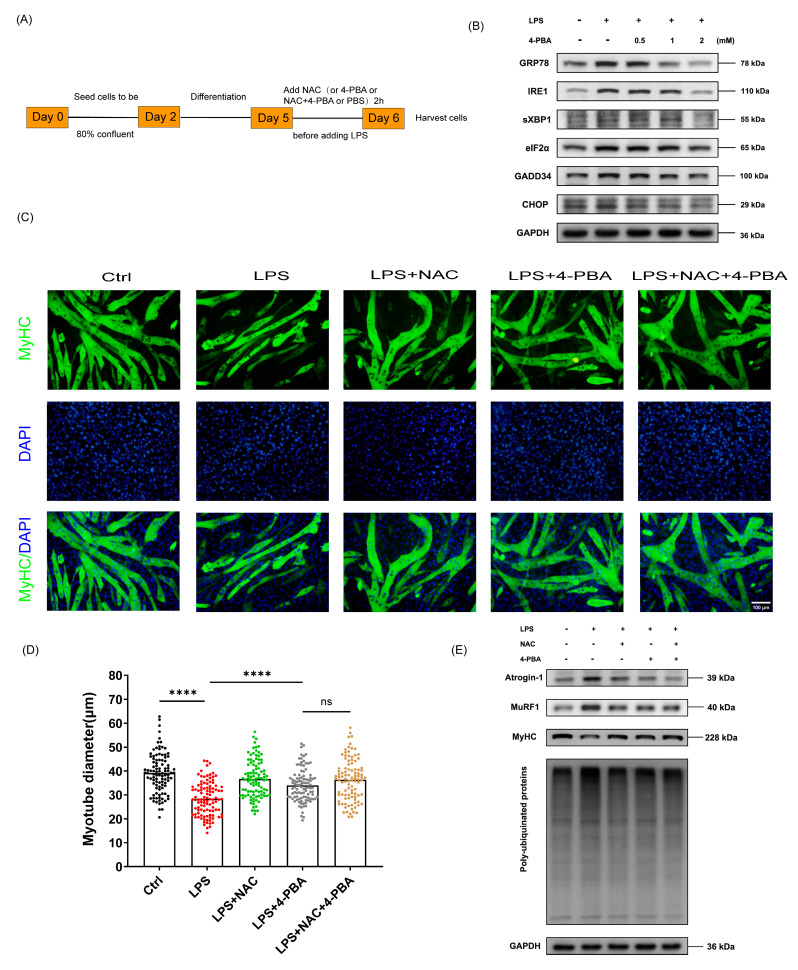
The protective effect of NAC on sepsis-induced muscle atrophy is partly mediated by ER stress. (**A**) Experimental protocol. (**B**) Indicated molecules of ER stress were determined by Western blot (*n* = 3). 4-PBA was used as an ER stress inhibitor. (**C**) Representative immunofluorescent images of each group (*n* = 3). (**D**) Statistics of myotube diameters. Scale bar = 100 μm. (**E**–**I**) The expression and quantifications of Atrogin-1, MuRF1, MyHC, and poly-ubiquitinated proteins (*n* = 3). 4-PBA: 4-Phenylbutyric acid. Black: control group; Red: LPS group; Green: LPS + NAC group; Grey: LPS + 4-PBA group; Yellow: LPS + NAC + 4-PBA group. * *p* < 0.05, ** *p* < 0.01, *** *p* < 0.001, and **** *p* < 0.0001 represent a significant difference. ns = not statistically significant.

**Figure 5 biomedicines-12-00902-f005:**
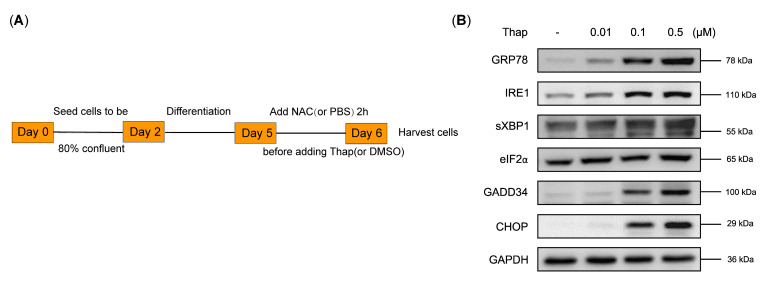
N–acetylcysteine attenuates ER stress-induced myotube atrophy. (**A**) Experimental protocol of this section. (**B**) Expression of ER stress biomarkers at varied concentrations of Thap (0.01, 0.1, 0.5 μM) (*n* = 3). (**C**) Immunofluorescence images of C2C12 myotubes treated with Thap with or without NAC (*n* = 3). Scale bar = 100 μm. (**D**) Measurement of myotube diameter after stimulation of Thap with or without NAC treatment. (**E**) Western blot of Atrogin-1, MuRF1, MyHC, and poly-ubiquitinated proteins after Thap treatment with or without NAC pretreatment (*n* = 3). (**F**–**I**) Quantifications of A. Thap: Thaspigargin. Black: control group; Red: Thapsigargin group; Blue: Thapsigargin + NAC (0.1 mM) group; Green: Thapsigargin + NAC (1 mM) group. * *p* < 0.05, ** *p* < 0.01, *** *p* < 0.001, and **** *p* < 0.0001 represent a significant difference.

## Data Availability

Data are contained within the article and Appendix A.

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
