# Peer review of "N-Acetylcysteine Attenuates Sepsis-Induced Muscle Atrophy by Downregulating Endoplasmic Reticulum Stress"

_biomedicines, 2024, doi:10.3390/biomedicines12040902_

Round 1

Reviewer 1 Report

Comments and Suggestions for Authors

The authors showed the protective effects of NAC in LPC-induced muscle atrophy. Measurements of muscle morphology and size, and western blot of a number of crucial proteins were performed.

However, the paper had the following problems:

1) Lack of controls, ie., in almost all the figures and parameters, the effects of NAC or 4-PBA alone, were not measured. Could it be that NAC itself just affecting basal values, or preventing the effects of LPS? The effects of NAC alone must be measured.

2) In each figure, the authors did not mention the numbers of independent experiments performed, albeit there were statistical significances.

3) In almost all the figures, the fonts are very small and thus blurry even I magnified them, that made reading painful and impossible.

Comments on the Quality of English Language

OK

Author Response

We would like to express our sincere gratitude to you for taking the time to review this manuscript. Please find the detailed responses below and the corresponding highlighted changes in the re-submitted files.We appreciate the positive comments from the editor and reviewers. According to the editor and the reviewer’s suggestions, we have completed the necessary revision of the manuscript, and included a point-by-point response to the reviewers’ critiques.

Reviewer 2 Report

Comments and Suggestions for Authors

The authors previous study has reported that ER stress promotes sepsis-induced muscle atrophy (17). It was shown that N-acetylcysteine (NAC) prevented skeletal muscle atrophy in diabetes (24). In this manuscript, the authors investigated the effect of NAC on sepsis-induced muscle atrophy using sepsis modeling in mice (cecal ligation and puncture-operated (CLP)) and LPS-treated C2C12 myotubes.

They showed that NAC prevents sepsis-induced increasing E3 ubiquitin ligases and muscle atrophy in vivo (figure 1), NAC attenuates LPS-induced myotube atrophy in vitro (figure 2), NAC attenuated several ER stress biomarkers in vivo and in vitro (figure 3), concomitant administration of NAC and 4-PBA, an ER stress pathway inhibitor, restored myotube diameter and the reduction of E3 ubiquitin ligases in LPS-treated myotubes but was not statistically different from 4-PBA alone in C2C12 myotubes (figure 4), and NAC restored Thapsigargin-induced myotube atrophy (figure 5).

Based on the above results they propose that the anti-muscle atrophy effect of NAC on sepsis was associated with suppressing ER stress pathway and ubiquitin–proteasome system (UPS) as well.

Although the exact mechanisms of NAC in suppressing ER stress pathway and UPS are obscured in this experiment, I think this work is well organized and makes a nice contribution to the understanding of the role of NAC in prevention of sepsis-induced muscle atrophy.

I have some concerns to address as below.

1.    The graph legends of all figures are blurry and difficult to read, it need to be replaced with higher quality versions.

2.    Are there any effects of NAC on body weight or tibialis anterior (TA) muscle size on Sham-operated muscle in figure 1?

3.    They show that 0.1,0.5,1mM of NAC cultured with cells for 24h had no significant difference in the viability of cells compared to control group (figure 2B). Again, are there any effects of NAC on myotubes in diameter on controlled C2C12 myotubes in figure 2?

Author Response

(The authors gave the same response as above.)

Reviewer 3 Report

Comments and Suggestions for Authors

Dear the Editor

Chen R et al reported a protective effect of NAC on sepsis-induced muscle atrophy in vivo and in vitro. First, these authors demonstrated an attenuated effect of NAC in cecum ligation-mediated sepsis model (Fig. 1). Then, this biochemical mechanism was further examined in C2C12 myotube model subsequently. Specifically, these authors observed an accumulation of ubiquitinated proteins and ER stress-induced proteins such as IRE1 and sXBP1 (Figs 3-5). In the last part of study, these authors further examined the role of 4-PBA, an ER stress pathway inhibitor and Thap, an agonist of ER stress, in this sepsis model (Figs 4&5). The aim of this study appeared clear with reasonable conclusions.

Minor concerns:
1) Manuscript did not appear to be carefully prepared. For example, 4-PBA (L371) and Thap (L380) were defined here, however these appeared many times in this manuscript.
2) All location and country of supplier need to be described in Section 2.
3) Fig 1A did not seem to be defined in text.
4) In LL170-172, no human study was involved in this study. 

Author Response

(The authors gave the same response as above.)

Reviewer 4 Report

Comments and Suggestions for Authors

There  are corresponding findings on muscle cells' classical histology, such as interstitial edema, cellular swelling, nuclear pyknosis, etc.?  On the other hand, there are acute microvascular lesions, which cause secondary conditions , such as hypoxia, acidosis, decreased glucose supply, responsible for subsequent myocells damage?

Author Response

(The authors gave the same response as above.)

Round 2

Reviewer 1 Report

Comments and Suggestions for Authors

revision is satisfactory

Reviewer 2 Report

Comments and Suggestions for Authors

In my opinion, in this revised manuscript, the responses to the major points raised previously were adequate and now this is acceptable for publication in Biomedicines.

Reviewer 3 Report

Comments and Suggestions for Authors

Dear the Editor

All raised concerns by this Reviewer were properly addressed by this revision.